# COVID-19 Phenotypes and Comorbidity: A Data-Driven, Pattern Recognition Approach Using National Representative Data from the United States

**DOI:** 10.3390/ijerph19084630

**Published:** 2022-04-12

**Authors:** George D. Vavougios, Vasileios T. Stavrou, Christoforos Konstantatos, Pavlos-Christoforos Sinigalias, Sotirios G. Zarogiannis, Konstantinos Kolomvatsos, George Stamoulis, Konstantinos I. Gourgoulianis

**Affiliations:** 1Department of Neurology, University of Cyprus, 75 Kallipoleos Street, Lefkosia 1678, Cyprus; 2Laboratory of Cardio-Pulmonary Testing and Pulmonary Rehabilitation, Department of Respiratory Medicine, Faculty of Medicine, University of Thessaly, Biopolis, 41500 Larissa, Greece; kgourg@uth.gr; 3Department of Respiratory Medicine, Faculty of Medicine, School of Health Sciences, University of Thessaly, Biopolis, 41500 Larissa, Greece; szarog@med.uth.gr; 4Department of Business Administration, University of Patras, University Campus—Rio, 26504 Patras, Greece; ckonstanta@upatras.gr; 5Department of Mechanical Engineering and Aeronautics, University of Patras, 26504 Patras, Greece; sinpack@gmail.com; 6Department of Physiology, Faculty of Medicine, School of Health Sciences, University of Thessaly, Biopois, 41500 Larissa, Greece; 7Department of Electrical and Computer Engineering, University of Thessaly, 37 Glavani—28th October Str., Deligiorgi Building, 4th Floor, 38221 Volos, Greece; kostasks@cs.uth.gr (K.K.); georges@e-ce.uth.gr (G.S.)

**Keywords:** COVID-19, pattern recognition, phenotypes, epidemiology, comorbidity, big data

## Abstract

The aim of our study was to determine COVID-19 syndromic phenotypes in a data-driven manner using the survey results based on survey results from Carnegie Mellon University’s Delphi Group. Monthly survey results (>1 million responders per month; 320,326 responders with a certain COVID-19 test status and disease duration <30 days were included in this study) were used sequentially in identifying and validating COVID-19 syndromic phenotypes. Logistic Regression-weighted multiple correspondence analysis (LRW-MCA) was used as a preprocessing procedure, in order to weigh and transform symptoms recorded by the survey to eigenspace coordinates, capturing a total variance of >75%. These scores, along with symptom duration, were subsequently used by the Two Step Clustering algorithm to produce symptom clusters. Post-hoc logistic regression models adjusting for age, gender, and comorbidities and confirmatory linear principal components analyses were used to further explore the data. Model creation, based on August’s 66,165 included responders, was subsequently validated in data from March–December 2020. Five validated COVID-19 syndromes were identified in August: 1. Afebrile (0%), Non-Coughing (0%), Oligosymptomatic (ANCOS); 2. Febrile (100%) Multisymptomatic (FMS); 3. Afebrile (0%) Coughing (100%) Oligosymptomatic (ACOS); 4. Oligosymptomatic with additional self-described symptoms (100%; OSDS); 5. Olfaction/Gustatory Impairment Predominant (100%; OGIP). Our findings indicate that the COVID-19 spectrum may be undetectable when applying current disease definitions focusing on respiratory symptoms alone.

## 1. Introduction

Since its emergence, COVID-19 has conceptually evolved from a viral pneumonia to a multisystem disease with insidious onset and diverse outcomes [1]. As additional cases caused a shift in case definitions, big data, and detailed symptom indexing arose as a necessity toward guiding evidence-based medicine and preventing severe outcomes [2]. An intrinsic perturbation in using expert-based definitions is inherent bias (i.e., as this is a hypothesis- or observation-driven approach), and an expected lack of recognition of fringe cases or spectrums that may however be deemed as such due to their underrepresentation within a given cohort.

Conversely, data-driven disease phenotyping aims to identify latent structures of potential or established disease descriptors within a given cohort; subsequently, the structures are scrutinized based their salient and unique characteristics in order to determine their characterization as true phenotypes [3]. This has been adopted by other research groups in obstructive sleep apnea, revealing that, i.e., severity based definitions could ignore non-linear relationships between disease characteristics and severity indices [3], as well as omit specific relationships (i.e., latent structures) between clinical manifestations and laboratory findings [4,5,6]. Other implementations in sleep-disordered breathing have shown that aside from classical definitions, i.e., based on an index of morbidity, different combinations of symptoms or laboratory findings could be used as phenotyping variables [7,8]. Aside from sleep-disordered breathing, this proposed methodology [3] has been used successfully in other disease models and implemented in other disease models, enabling the ad hoc development of diverse phenotyping concepts in idiopathic Parkinson’s disease, obesity, and venous thromboembolism [9,10,11,12]. Furthermore, data from multicenter studies have also been shown produce biologically relevant phenotypes via our proposed methodology [13]. These studies have shown that a data-driven, pattern recognition approach is both flexible, robust, and allows the discovery of phenotypes that are independent of clinical preconceptions, which are often subject to salience bias. 

We hypothesized that data-driven recognition of COVID-19 phenotypes will allow an unbiased mapping of the global clinical spectrum. Furthermore, these data-driven phenotypes would enable the design of clinical studies and enhance outcome design and evaluation, producing efficacious, bias-free treatments and healthcare policy interventions.

Therefore, the specific aims of the study were the following: The use of reported symptoms to identify latent structures via categorical PCA and dimension reduction approaches, using data from the COVID-19 Delphi Facebook study.To scrutinize the previously created latent structures as potential COVID-19 phenotypes or phenotyping parameters via TSC and artificial intelligence-based classification.

## 2. Materials and Methods

### 2.1. Study Population

Data for this study were extracted from the CTIS Trends and Impact Survey (CTIS), based on symptom surveys developed by the Delphi group at Carnegie Mellon University (CMU). Initially, Facebook selects a random sample among its users in the United States. The users are then presented with the option to participate in the study. In turn, participation entails the administration of the surveys’ iteration, and covers data on COVID-19-like symptoms, behavioral, mental health, and economic parameters, as well as estimates the impact of the pandemic on the responders daily life. Individual, anonymized survey responses are stored in CMU’s servers and made accessible to healthcare professionals under a project-specific data use agreement. Approximately 50,000 responders participate in the study per day, with monthly survey results comprising more than 1 million responders per month.

A detailed overview of the CTIS, its conception, and evolution are available from: https://delphi.cmu.edu/covid19/ctis/ (accessed on 25 April 2020).

### 2.2. Study Design

We performed a retrospective analysis of CTIS data collected between May to December in this study, and we included responders with a certain COVID-19 status, i.e., having answered either “Yes” or “No” in the corresponding item of the survey.

An indicative item structure from Wave 4 of the study is the following (note that B10 and B10a are the codes for the corresponding questionnaire items):

B10—Have you been tested for coronavirus (COVID-19) in the last 14 days?

B10a—Did this test find that you had coronavirus (COVID-19)?

A positive COVID-19 status was assigned to responders answering “Yes” in B10a, whereas a negative COVID-19 status was assigned to responders answering “No” in the same item. Responders that answered “I do not know” in item B10a were excluded from further analyses.

As a subsequent exclusion criterion, we employed a cutoff of ≤30 days in symptom duration. This cutoff was selected on the premises of an approximate “return to wellness”, estimated to occur at 14–21 days for 65% of patients a positive outpatient test result, in a recent report by the CDC [14]. The purpose of this cut-off was to simultaneously include COVID-19 patients with longer durations of illness and to exclude symptom durations unlikely to be attributable to COVID-19 manifestations, such as 60 days or more. In order for our approach to be forward and backward compatible, we opted to use the very first incarnation of symptoms attributed to COVID-19 and captured by the survey (i.e., the first wave). As such, a core of the 13 first symptoms recorded by the survey would remain the same, regardless of future additions. 

### 2.3. Statistical Analysis

#### Pattern Recognition via Multiple Correspondence Analysis

Symptom data were used by combining a logistic regression-based case scoring (See Appendix A) dimension reduction technique and cluster analysis algorithm, as previously described [9,15]. Specific COVID-19 symptoms, encoded as survey items, were used as input variables for multiple correspondence analysis (MCA). In turn, MCA derived object scores for each case, which were subsequently used as input variables for the cluster analysis, along with symptom duration of COVID-19 positive responders [9,16]. The optimal number of MCA-derived dimensions was determined based on achieving a total variance (i.e., cumulative variance per dimension) of >70% [17]. MCA and MCA preprocessing prior to cluster analysis are techniques that allow the identification of latent patterns within a population, based on a set of nominal response variables [18]; OR-weighting of the input variables was used here as a ranking scheme, based on their association with COVID-19-positive responders vs. COVID-19-negative responders.

### 2.4. Two-Step Clustering and Phenotype Extraction

OR-weighted MCA-produced case-wise object scores (i.e., composite quantifications of symptoms per case) along with symptom durations were subsequently used by the Two Step Clustering (TSC) algorithm to produce symptom phenotypes. Initially, Two Step Clustering merges raw input data into primary subclusters. The second step employs a hierarchical clustering method that aims to merge the subclusters into progressively larger clusters. This process does not require the a priori determination of a set number of clusters. As we and others have previously demonstrated, TSC is well suited for the identification of latent phenotypes in a given population [9,18]. In this study, the Log-likelihood was used as a distance measure, and the Bayesian Information Criterion (BIC) was used as the clustering criterion for the automatic determination of cluster number.

### 2.5. Phenotype Validation: Cross-Sectional and Longitudinal Aspects

The model that was created on the pilot analysis of 66.165 responders within the August dataset, and was subsequently validated in monthly data ranging from March–December 2020. Specifically, the procedure was as follows:(a)The weights extracted from August’s responders were applied to an MCA based on symptom data recorded for each subsequent and preceding month’s responders.(b)Object scores were calculated for each responder.(c)Object scores and symptom duration per month were used in TSC.

### 2.6. Crossectional Validation: Phenotypes vs. Controls

Cross sectional validation of the produced phenotypes essentially answers the question of whether a COVID-19 syndrome is associated with a positive COVID-19 test. For this purpose, Receiver Operator Characteristic (ROC) curves were used to determine the diagnostic accuracy of a symptom-based probability for each phenotype when compared to controls. Specifically, ROC curves were fitted by the probability pi, extracted from the application of the logistic regression model derived from August’s data. Hence, pi would be expressed as follows:(1)$$p_{i}=\frac{e^{a_{0}+a_{1}i_{1}+a_{2}i_{2}+\ldots a_{n}i_{n}}}{1+e^{a_{0}+a_{1}i_{1}+a_{2}i_{2}+\ldots a_{n}i_{n}}}\;,\;0<p<1$$
where i_1_, i_2_, …, i_n_ is the symptom per i-th month, and a_0_, a_1_, …, a_n_ are the f extracted from August’s dataset. This computed probability P_i_ of cluster membership was used as an input variable for ROC curve fitting. Finally, COVID-19 status was used as a binary dependent variable for the ROC curve, and the area under curve (AUC) was calculated per month, for each cluster.

### 2.7. Longitudinal Validation: Phenotype Re-Emergence and Symptom Invariance

The primary criterion for validation was the emergence of consistent phenotypes in at least one month other than August, based on complete or quasi-complete symptom separation per phenotype. Essentially, this would translate to the identification of each phenotype based on the most salient symptoms.

The secondary criterion was based on the hypothesis that re-emergent phenotypes would be furthermore identified based on non-salient, non-preclusive symptoms. To meet this criterion, frequency tables for each symptom were constructed per each month and phenotype. 

For each symptom S reported on each month M, for a number of months N, we consider the mean, μ_s_:(2)$$\mu_{s}=\frac{S_{1}+S_{2}+\ldots S_{N}}{N}$$

In order to assess symptom perseverance and their non-random contribution as patterns within each phenotype, we perform a normality test under the null hypothesis that the distribution of a symptom/month [S_1_, S_2_, …, S_N_] is normal, and therefore 95% of the observations lie within two standard deviations of μ_s_. The following concept has been previously used in face recognition algorithms [19]. Here, we used the Shapiro–Wilk test of normality, and a *p*-value < 0.05 was considered statistically significant. Symptoms achieving below threshold p-values were considered variable for each corresponding phenotype. Correspondingly, longitudinal symptom invariance (SI) is described as follows:(3)$$\mathrm{SI}=1-\frac{S_{v}}{M}$$
where S_v_ is the number of variant symptoms (defined by a Shapiro–Wilk *p*-value < 0.05) and M is the total number of symptoms. SI is equal to 1 (100%) when S_(v)_ = 0 and SI is equal to 0 (0%) when S_(v)_ = M. 

### 2.8. Post-Hoc Analyses 

Associations between each phenotype and symptoms were determined via a combination of the χ^2^ test with adjusted standardized residuals (Standardized Pearson residuals). A χ^2^ test *p*-value < 0.05 and adjusted standardized residuals either greater than 1.96 or less than -1.96 was considered statistically significant. Associations among phenotypes and responder demographic and medical history characteristics (age group, gender, and comorbidity) were investigated via a logistic regression model. 

### 2.9. Determination of Data-Driven Diagnostic Rules via Decision Tree Analyses

For each of the symptom-based phenotypes, certain symptoms were obligatory, i.e., characterized 100% for this purpose; we performed Decision Tree Analyses (DTA) via the Quick, Unbiased, Efficient, Statistical Tree (QUEST) algorithm [20]. Decision tree analysis is a data mining technique that is implemented in order to create a classification scheme from a set of observations; in biomedical research, its main applications include the creation of data-driven diagnostic or predictive rules [21,22]. A detailed presentation of the method is included in Appendix A.

Each *p*-value < 0.05 was considered statistically significant. All analyses were performed SPSS version 24.0 (IBM, Chicago, IL, USA). 

## 3. Results

### 3.1. Study Population

The total study population included 320,326 responders with a certain COVID-19 test status and disease duration < 30 days (Figure 1). Table 1 presents the study population’s demographics per month. Appendix A presents multinomial regression results per month and phenotype (A1–A5). Figure 2 presents temporal relationships between phenotypes and symptoms, i.e., whether clusters identified in August re-emerged in preceding and succeeding months. Based on our approach, rose charts can effectively visualize the (i) symptom invariance criterion and (ii) symptom pattern re-emergence, both of which were required for the recognition of phenotype stability from concept (August) to validation (preceding and succeeding months).

### 3.2. Phenotype Extraction

Based on the latent structures between symptom data and disease duration, five COVID-19 syndromes (Figure 2; Table 2) were extracted from August’s 61,165 responders:Afebrile (0%), Non-Coughing (0%), Oligosymptomatic (ANCOS).Febrile (100%) Multisymptomatic (FMS).Afebrile (0%) Coughing (100%) Oligosymptomatic (ACOS).Oligosymptomatic with additional self-described symptoms (100%; OSDS).Olfaction/Gustatory Impairment Predominant (100%; OGIP).

Validation and further characterization

Repeating the multiple correspondence and cluster analyses per subsequent and preceding month (April–December), resulted in the validation of each phenotype as follows:(a)ANCOS and OSDS emerged in 10/10 months(b)MFS and ACOS emerged in 9/10 months(c)OGIP emerged in 4/10 months.

Based on the most salient symptoms, decision trees were subsequently constructed (Figure 3), providing a structured approach in identifying each phenotype. 

Further characterization of these five phenotypes was achieved via identifying invariant symptoms between April–December (Figure 2); based on these observations and the results of the Shapiro–Wilk tests:(a)ANCOS was characterized by general malaise in the absence of fever and upper respiratory tract symptoms.(b)ACOS was characterized as a mainly afebrile upper respiratory tract viral infection.(c)FMS was a more typical, febrile syndrome covering respiratory and gastrointestinal (GI) manifestations.(d)OGIP, the most invariant syndrome, was characterized by the absence of fever and diarrhea.(e)OSDS did not typically include symptoms of pain or pressure on the chest, nor difficulty in breathing.

Interestingly, the implementation of “Headache” in December as a standalone question resulted in a decomposition of the OSDS phenotype. This is further exemplified by the comparison between text-mined headache as a symptom in August (10% of OSDS) versus a three-fold increase in prevalence when asked directly in December.

Multiple nominal regression of comorbidities, adjusted for age group and gender, revealed several statistically significant associations (Appendix A). Notably, a history of asthma and chronic lung disease were abortive comorbidities for certain phenotypes (ANCOS, ACOS, OGIP for asthma, and additionally OSDS for chronic lung disease). Gender and age group did not display sequentially consistent associations with any phenotype.

## 4. Discussion

In our study, five distinct COVID-19 phenotypes were identified: (a) Afebrile (0%), Non-Coughing (0%), Oligosymptomatic (ANCOS); (b) Febrile (100%) Multisymptomatic (FMS); (c) Afebrile (0%) Coughing (100%) Oligosymptomatic (ACOS); (d) Oligosymptomatic with additional self-described symptoms (100%; OSDS); (e) Olfaction/Gustatory Impairment Predominant (100%; OGIP). Validation of these phenotypes revealed that, based on symptom pattern re-emergence: (a) ANCOS and OSDS emerged in 10/10 months, (b) MFS and ACOS emerged in 9/10 months, (c) OGIP emerged in 4/10 months. The symptom invariance criterion revealed that, between April–December: (a) ANCOS was characterized by general malaise in the absence of fever and upper respiratory tract symptoms, (b) ACOS was characterized as a mainly afebrile upper respiratory tract viral infection, (c) FMS was a more typical, febrile syndrome covering respiratory and gastrointestinal (GI) manifestations, (d) OGIP, the most invariant syndrome, was characterized by the absence of fever and diarrhea, and (e) OSDS did not typically include symptoms of pain or pressure on the chest, nor difficulty in breathing. Additionally, direct inquiry for headache as a symptom in December resulted in a decomposition of the OSDS phenotype. This is further exemplified by the comparison between text-mined headache as a symptom in August (10% of OSDS) versus a three-fold increase in prevalence when asked directly in December. Multiple nominal regression of comorbidities, adjusted for age group and gender, revealed that asthma and chronic lung disease were abortive comorbidities for certain phenotypes (ANCOS, ACOS, OGIP for asthma, and additionally OSDS for chronic lung disease). 

The presence or absence of fever, cough, olfactory/gustatory dysfunction, and atypical symptoms defined these phenotypes as their primary features. The concept of symptom invariance was subsequently used to further determine their stability regarding symptom composition, indicating that the olfactory/gustatory predominant phenotype OGIP was the most invariant, i.e., the most stable, across the 4 months that it emerged in. Finally, while several comorbidities such as heart disease and diabetes were associated with the risk of manifesting specific phenotypes, other comorbidities such as asthma were found to be abortive. 

After its initial identification as a novel pneumonia, the increasing numbers of COVID-19 cases began to outline a spectrum [23], rather than a linear progression from mild viral infection to a severe one [24]. The recognition of COVID-19′s heterogeneity however was initially limited within the setting of treatment response or severity phenotypes [25,26], while the heterogeneity of non-severe cases or those lacking a salient respiratory aspect was not addressed. Even within the concept of point care phenotyping however, phenotypes similar to those identified in our study have been described by independent studies. Bayesian approaches have identified phenotypes corresponding to FMS, OGIP, and ACOS in the clinical setting, and have included them in diagnostic algorithms [27].

In our cohort, a similar diagnostic rule emerges and recurs in monthly aggregated data, corresponding to the phenotypes identified here (Figure 2).

The importance of phenotyping COVID-19 outside the initially severe or point of care spectrum becomes evident when examining previous iterations of diagnostic criteria; fever and respiratory symptoms were initially the only manifestations considered relevant in defining cases [28]. One of the largest studies on initially asymptomatic or non-respiratory symptom (NRS) phenotype of COVID-19 patients has shown that this approach may miss a portion of active cases that may subsequently convert to severe manifestations [29]. Our findings support this concept, with NRS overlapping with the OGIP, OSFS, and ANCOS phenotypes.

As previous research has suggested, comorbidities were found to be independent predictors of COVID-19 phenotypes, even after adjustments for age group and gender (Appendix A). As a general rule, two broad categorizations of comorbidities can be inferred: those that can intertwine with the pathophysiology of SARS-CoV-2, such as diabetes [30] and heart disease [31], and those where a treatment effect may restrict phenotype manifestations.

In this light, several noteworthy associations include comorbid asthma and chronic lung disease, which appear to reduce the risk of manifesting the FMS phenotype (Appendix A). This seemingly paradoxical relationship has been previously explored in the literature, and mainly attributed to the protective effects of inhaled corticosteroids (ICS); Specifically, while their use may lead to quiescent type I/III interferon responses, the concomitant downregulation of ACE2 and TMPRSS2 may restrict SARS-CoV-2 from entering pneumonocytes [32,33]. As our data are limited regarding medication use and the specific respiratory disease of each responder, we cannot safely attribute the associations observed to ICS usage.

Based on current literature, ICS treatment plausibly presents a potential phenotype abortive effect in otherwise vulnerable populations such as asthma patients [34]. Recent evidence on the efficacy of ICS as ad hoc COVID-19 treatments provide further support for this concept [35].

In a similar fashion, the phenotype abortive effect of cancer as a comorbidity could reflect yet another treatment rather than disease effect. Such an effect may account for this association, considering that several anti-cancer treatments are undergoing trials as repurposed COVID-19 treatments [36]. As no data were collected on primary tumors, staging, or treatment [37], due to the nature of the survey, this hypothesis cannot be scrutinized further.

### Limitations and Strengths

The results of our study should be interpreted within the context of their limitations. As a survey administered via Facebook, our source data incur the corresponding selection bias. This however is potentially balanced by the large sample size of the final cohort, and represents the single largest study of its kind. Survivor bias is also inherently present in our study, considering that responders are unlikely to have had severe COVID-19 at the time of survey administration. The lack of follow-up data correspondingly precludes that phenotype shifts (e.g., ANCOS or OSDS to FMS) cannot be explored. Another important consideration is that OSDS inevitably absorbs symptoms not originally covered by the initial study iterations and is correspondingly decomposed when these symptoms are identified and added. A prime example of this case is headache as symptom; when left to the discretion of the responder, it might not be evaluated properly as a feature [36]. This paradigm becomes evident by the discordance between text-mining (April–November data, 10% of OSDS) vs. being asked directly (i.e., 30% in all phenotypes and decomposition of OSDS as a “pure” phenotype). Comorbidities, reported by majority as broad categories, cannot be safely considered in strict interpretations as to their associations with phenotypes. Finally, as gender categories beyond male/female are underrepresented in the monthly samples, they cannot be safely used to extrapolate their contribution on clinical phenotypes. These intrinsic caveats of the study are inherited from the broader structure of the data, and the post-hoc extraction of a data subset for a specific concept (i.e., data driven phenotyping of COVID-19 syndromes).

## 5. Conclusions

The main strength of the study was the determination and retro- and anterograde validation of COVID-19 syndromes in the largest community sample to date. The phenotypes we uncovered solidify phenotypes previously described by independent studies, and furthermore provide the basis and tools for the development of utilizable diagnostic rules. One of the most important concepts explored here is that the febrile respiratory phenotype represents a lesser portion of COVID-19 phenotypes in the community, a finding that should be considered both in epidemiological profiling and healthcare provision. Our findings support the concept of symptom-based phenotypes of COVID-19 that remain distinct within 9–12 days from first symptom onset. The existence of phenotypes rather than severity strata may further explain the low diagnostic accuracy achieved by rule in or rule out algorithms based solely on symptoms, without accounting for their dependency and intercorrelations, even between different systems (i.e., GI and respiratory). In order to utilize our findings in the clinical setting and make them available to other researchers, we have developed an online application that calculates the symptom-based logistic probability Px (Available from: http://se8ec.csb.app, (accessed on 24 April 2020)). In this community-based sample, febrile respiratory disease was infrequent when compared to atypical presentations within a range of 9–12 days from symptom onset; this finding may be critical in current epidemiological surveillance and the development of transmission dynamics concepts.

## Figures and Tables

**Figure 1 ijerph-19-04630-f001:**
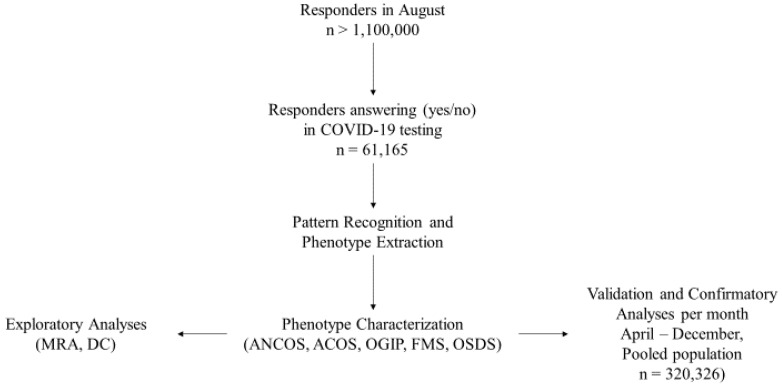
Study Workflow. Phenotype legends: 1. Afebrile (0%), Non-Coughing (0%), Oligosymptomatic (ANCOS); 2. Febrile (100%) Multisymptomatic (FMS); 3. Afebrile (0%) Coughing (100%) Oligosymptomatic (ACOS); 4. Oligosymptomatic with additional self-described symptoms (100%; OSDS); 5. Olfaction/Gustatory Impairment Predominant (100%; OGIP).

**Figure 2 ijerph-19-04630-f002:**
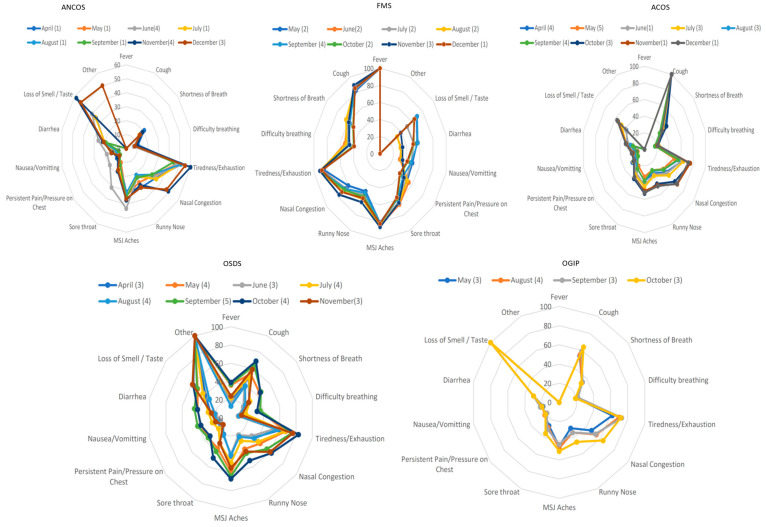
Rose charts presenting the temporal relationships between phenotypes and symptoms. Phenotype legends: 1. Afebrile (0%), Non-Coughing (0%), Oligosymptomatic (ANCOS); 2. Febrile (100%) Multisymptomatic (FMS); 3. Afebrile (0%) Coughing (100%) Oligosymptomatic (ACOS); 4. Oligosymptomatic with additional self-described symptoms (100%; OSDS); 5. Olfaction/Gustatory Impairment Predominant (100%; OGIP). The numbers in parentheses represent the cluster numbers from the validation process, e.g., cluster number n in August and cluster number c in June would both correspond to FMS.

**Figure 3 ijerph-19-04630-f003:**
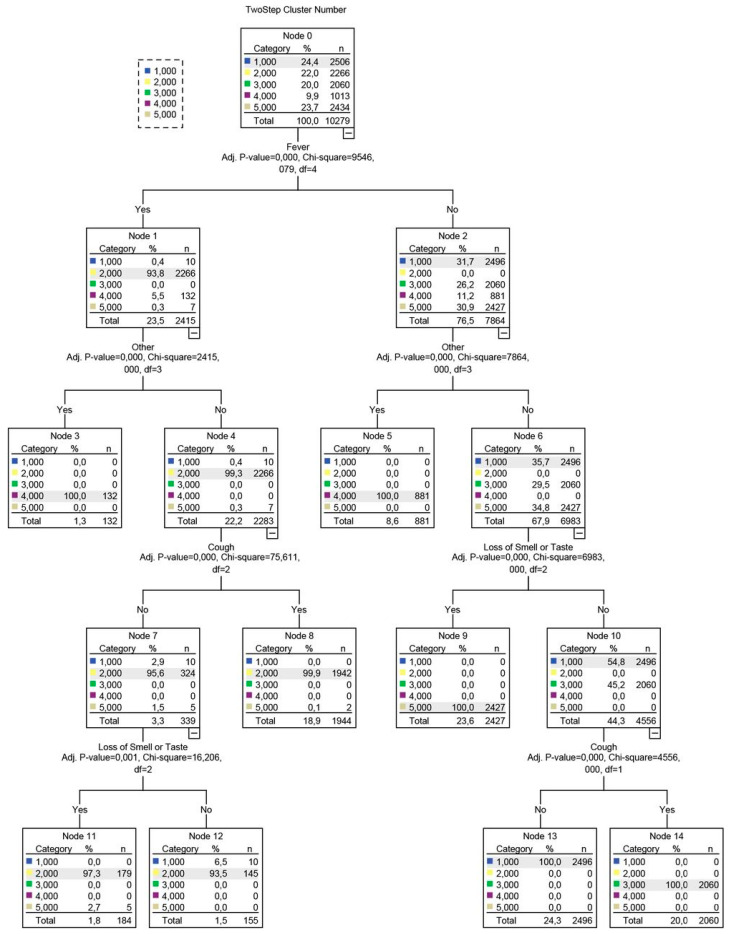
Decision tree developed using the QUEST algorithm. The decision tree’s branches are based on splits, i.e., variables that were selected based on a Chi-squared test-determined *p*-value. The dependent variable for this analysis was a cluster number, represented as a nominal categorical variable with 5 levels, corresponding to the initial 5 clusters detected in August.

**Table 1 ijerph-19-04630-t001:** Demographics per month of the study population.

		April (*n* = 22,320)	May (*n* = 38,043)	June (*n* = 51,582)	July (*n* = 78,951)	August (*n* = 66,155)	September (*n* = 12,801)	October (*n* = 19,137)	November (*n* = 22,698)	December (*n* = 48,629)
COVID-19		4000	4955	6573	13,370	10,279	1773	5936	10,026	21,617
Age Group	18–24	1783	2973	4202	7047	5965	1194	1211	1606	3312
	25–34	4856	7423	9837	15,559	12,158	2204	3353	4805	9002
	35–44	4794	7249	8919	14,450	11,502	2116	3886	4856	9529
	45–54	4281	7030	8886	13,660	11,146	2119	3519	4095	8809
	55–64	3220	6235	8661	12,385	10,775	2173	3320	3227	7440
	65–74	1447	3597	5655	7791	7218	1483	1808	1539	3907
	>75	312	907	1632	2317	2155	531	573	427	1181
	NA	1627	2629	3790	5742	5246	981	1467	2143	5449
Gender	M	5001	9754	13,230	20,313	17,427	3434	4555	4752	10,684
	F	15,459	24,985	33,672	51,556	42,364	8170	12,732	15,360	31,612
	NB	149	284	366	613	532	104	141	176	347
	SD	102	219	252	357	297	72	119	144	248
	NA	153	325	449	653	565	100	128	128	322
	N/A	1456	2486	3613	5459	4980	921	1462	2138	5416
Cancer		1223	2338	3209	4353	3783	816	1082	1040	2289
	HD	6556	7768	8450	13,352	10,792	2036	4738	6213	12,229
	HTN	3952	4682	5129	8263	6486	1302	2979	3931	7785
Asthma		12,222	19,035	24,962	38,645	32,205	6031	11,461	14,309	28,597
	CLD	9361	14,946	19,201	30,197	25,728	5141	9644	12,504	25,272
	KD	8035	12,264	14,668	22,830	19,393	3907	8203	10,029	20,615
	AD	8807	14,702	19,365	28,438	23,692	4587	8450	10,533	21,677
Diabetes	T1D	4817	8313	9950	17,181	12,822	2627	5102	5565	12,807
	T2D	4202	4397	4556	7698	5899	1110	2803	3933	7743
	IC	3091	5150	5847	10,235	7325	14755	3519	4087	9238

Notes: Age Groups are measured in years. Cancer was specified as any form of neoplasm except skin cancer. AD: Autoimmune Disease; CLD: Chronic Lung Disease such as COPD; F: Female; HD: Heart Disease; HTN: Hypertension; IC: Immunocompromised. KD: Kidney Disease; M: Male; N/A: Not answered; NA: Not available; NB: Non-Binary; SD: Self-Described. Please note that for November’s responders we selected participants that received wave 4 of Delphi Study Questionnaire.

**Table 2 ijerph-19-04630-t002:** Cluster composition and symptom-based prediction vs. COVID-19—(controls)—August.

	N	AUC	*p*-Value	95% CI
ANCOS (1)	2506	<0.5	NA	NA
FMS (2)	2266	0.963	<0.001	0.961–0.965
ACOS (3)	2060	0.737	<0.001	0.729–0.746
OSDS (4)	1013	0.777	<0.001	0.762–0.792
OGIP (5)	2434	0.983	<0.001	0.982–0.984

Notes: Five COVID-19 syndromes were identified in August: 1. Afebrile (0%), Non-Coughing (0%), Oligosymptomatic (ANCOS); 2. Febrile (100%) Multisymptomatic (FMS); 3. Afebrile (0%) Coughing (100%) Oligosymptomatic (ACOS); 4. Oligosymptomatic with additional self-described symptoms (100%; OSDS); 5. Olfaction/Gustatory Impairment Predominant (100%; OGIP). AUC: Area Under Curve.

## Data Availability

All data are available upon request.

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
