# Peer review of "COVID-19 Phenotypes and Comorbidity: A Data-Driven, Pattern Recognition Approach Using National Representative Data from the United States"

_ijerph, 2022, doi:10.3390/ijerph19084630_

Round 1

Reviewer 1 Report

The goal is challenging and the idea is intriguing, but I find the manuscript cryptic, unstructured and not very reproducible. I suggest major revision.

Abstract

L23 using the survey results based on survey results from Carnegie Mellon University’s Delphi 23 Group

Better “using a Carnegie Mellon University’s Delphi Group survey-based cross sectional study”

Introduction

The introduction is sparse. Better describe the varied picture of COVID-19. Better describe the factor analysis approaches, background and future perspectives of these models

Methods

70 The design is missing. Is the database accessible? Have studies been carried out on this already?

eligibility of participants?

eligibility of participants should be detached from data processing. b10 what does it mean?

The data processing is unstructured, it is better to move the algorithms to supplementary material. Describe how you have performed the MCA with which variables described, if you have described the eigenvalues or plotted the Scree plot. How did you approach cluster analysis, with regression index, RMSE? as well as the ROCs.

I do not understand the connection between the MCA and the decision tree approach which is purely of a classification nature.

Results

Figure 2, I do not consider appropriate the rose charts.. why the coordinates of variable categories missing for MCA? Dimensions and eigenvalues?

Figure 3, what do the 5 clusters correspond to?

Discussion

At the beginning of the discussion paraphrase the study objective and then describe the major findings

Author Response

We thank the reviewer for the comment that have helped us to improve the paper. All changes have been indicated by red color within the text. Below you will find a point-by-point response to your comments.

Reviewer 1: Abstract: L23 using the survey results based on survey results from Carnegie Mellon University’s Delphi 23 Group. Better “using a Carnegie Mellon University’s Delphi Group survey-based cross sectional study”

Response: Thank you for your comment. Following the reviewer’s suggestion and after consulting with our collaborators in CMU, the references in the abstract and throughout the text have been amended to COVID-19 Trends and Impact Surveys (CTIS), which is the survey’s current (2022) official name.

Reviewer 2: Introduction: The introduction is sparse. Better describe the varied picture of COVID-19. Better describe the factor analysis approaches, background and future perspectives of these models

Response: Thank you for your comment. We have amended the introduction in order to include the studies mentioned in greater detail as they are applied in sleep disordered breathing as well as other disease models such as Parkinson’s disease, DVT and obesity, among others.

Reviewer 3: Methods: 70 The design is missing. Is the database accessible? Have studies been carried out on this already? eligibility of participants? eligibility of participants should be detached from data processing.

Response: Thank you for your comment. Eligibility here refers to whether data would be included based on our post-hoc criteria, not the initial enrolment criteria in CTIS.

Reviewer 4: b10 what does it mean?

Response: Thank you for your comment. B10 refers to the specific coding of the survey item in the original data provided  by the CTIS. We have also clarified this in-text.

Reviewer 5: The data processing is unstructured, it is better to move the algorithms to supplementary material. Describe how you have performed the MCA with which variables described, if you have described the eigenvalues or plotted the Scree plot. How did you approach cluster analysis, with regression index, RMSE? as well as the ROCs.

Response: Thank for your comment. Based on the reviewer’s suggestions we have clarified the MCA and its input process as follows: “Specific COVID-19 symptoms encoded as survey items, were used as input variables for multiple correspondence analysis (MCA). In turn, MCA derived object scores for each case, which were subsequently used as input variables for the cluster analysis, along with symptom duration on COVID-19 positive responders [9, 14].” Furthermore, we clarify that the optimal number of dimensions was based on cumulative variance rather than via fitting a Scree plot: The optimal number of MCA-derived dimensions was determined based on achieving a total variance (i.e. cumulative variance per dimension) of >70% [15].Finally, we clarify that TwoStep clustering follows a specific process: Initially, TwoStep clustering merges raw input data into primary subclusters. The second step employs a hierarchical clustering method that aims to merge the subclusters into progressively larger clusters. This process does not require the a priori determination of a set number of clusters. As we and others have previously demonstrated, TSC is well suited for the identification of latent phenotypes in a given population [9, 16]. In this study, the Log-likelihood was used as a distance measure, and the Bayesian In-formation Criterion (BIC) was used as the clustering criterion for the automatic determination of cluster number. And the ROC curve fitting as follows: This computed probability Pi of cluster membership was used an input variable for ROC curve fitting.

Reviewer 6: I do not understand the connection between the MCA and the decision tree approach which is purely of a classification nature.

Response: Thank you for your comment. We have amended the text to clarify the decision tree analysis’ purpose. Specifically, we indicate that based on the (a posteri) finding of salient symptoms among clusters, we attempted to create a diagnostic rule that would follow a “yes/no” approach. In essence, we attempted to predict whether a clinician looking for these symptoms could predict cluster membership. This analysis was independent of the analysis initial arm.

Reviewer 7: Results: Figure 2, I do not consider appropriate the rose charts…. why the coordinates of variable categories missing for MCA? Dimensions and eigenvalues?

Response: Thank you for your comment. Based on our approach, rose charts can effectively visualize the (i) symptom invariance criterion and (ii) symptom pattern re-emergence, both of which were required for the recognition of phenotype stability from concept (August) to validation (preceding and succeeding months). We have updated this rationale within text as per your suggestion: Figure 2 present temporal relationships between phenotypes and symptoms, i.e. whether clusters identified in August re-emerge in preceding and succeeding months. Based on our approach, rose charts can effectively visualize the (i) symptom invari-ance criterion and (ii) symptom pattern re-emergence, both of which were required for the recognition of phenotype stability from concept (August) to validation (preceding and succeeding months).

Response: Thank you for your comment. Based on (i) symptom invariance criterion and (ii) symptom pattern re-emergence criteria, we did not consider MCA dimensions and eigenvalues informative, as they were solely created in order to produce object scores per dimension, and not study symptom dimensionality or attempt a dimension reduction approach. Our previous work (1) and those of others (2) follows a similar presentation scheme that focuses on cluster characteristics and their validation as clinical phenotypes.

  1. Vavougios GD, George D G, Pastaka C, Zarogiannis SG, Gourgoulianis KI. Phenotypes of comorbidity in OSAS patients: combining categorical principal component analysis with cluster analysis. J Sleep Res. 2016 Feb;25(1):31-8. doi: 10.1111/jsr.12344.
  2. Shoji, Tomokazu et al. “Clinical Implication of the Relationship between Antimicrobial Resistance and Infection Control Activities in Japanese Hospitals: A Principal Component Analysis-Based Cluster Analysis.” Antibiotics (Basel, Switzerland) vol. 11,2 229. 10 Feb. 2022, doi:10.3390/antibiotics11020229

Reviewer 8: Figure 3, what do the 5 clusters correspond to?

Response: Thank you for your comment. The 5 clusters respond to August’s data, and amended in the figure description as follows: Decision Tree developed using the QUEST algorithm. The decision tree’s branches are based on splits, i.e. variables that were selected based on a chi-square test-determined p-value. The dependent variable for this analysis was a cluster number, represented as a nominal categorical variable with 5 levels corresponding to the initial 5 clusters detected in August.

Reviewer 9: Discussion: At the beginning of the discussion paraphrase the study objective and then describe the major findings

Response: Thank you for your comment. Based on the reviewer’s suggestion, we have added the following: In our study, five distinct COVID-19 phenotypes were identified: (a)     Afebrile (0%), Non-Coughing (0%), Oligosymptomatic (ANCOS),  (b)          Febrile (100%) Multi-symptomatic (FMS), (c) Afebrile (0%) Coughing (100%) Oligosymptomatic (ACOS) (d)                 Oligosymptomatic with additional self-described symptoms (100%; OSDS) and (e)               Olfaction / Gustatory Impairment Predominant (100%; OGIP). Validation of these phenotypes revealed that based on symptom pattern re-emergence: (a) AN-COS and OSDS emerged in 10/10 months, (b) MFS and ACOS emerged in 9/10 months, (c)         OGIP emerged in 4/10 months. The symptom invariance criterion revealed that, between April – December: (a) ANCOS was characterized by general malaise in the absence of fever and upper respiratory tract symptoms, (b)            ACOS was character-ized as a mainly afebrile upper respiratory tract viral infection, (c)          FMS was a more typical, febrile syndrome covering respiratory and gastrointestinal (GI) manifestations,  (d)          OGIP, the most invariant syndrome, was characterized by the absence of fever and diarrhea, (e)          OSDS did not typically include symptoms of pain or pressure on the chest, nor difficulty in breathing. Additionally, direct inquiry for headache as a symptom in December resulted in a de-composition of the OSDS phenotype. This is further exemplified by the comparison between text-mined headache as a symptom in August (10% of OSDS) versus a 3-fold increase in prevalence when asked directly in December. Multiple nominal regression of comorbidities, adjusted for age group and gender, revealed that asthma and chronic lung disease were abortive comorbidities for certain phenotypes (ANCOS, ACOS, OGIP for asthma and additionally OSDS for chronic lung disease).

Reviewer 2 Report

The article is well-written and interesting, giving statistical validation to clinical impressions of first-line healthcare workers in pandemia.

I will just suggest to turn this statement impersonal and focus more on explanation of the methodology. "We were the first to phenotype comorbidity in obstructive sleep apnea (OSA) using a 52
combination of preclustering principal component analysis to identify latent structures 53
within our datasets, and subsequently map them using the Two-Step Cluster algorithm 54
[3]. The methodology we proposed has been adopted by other research groups [4 - 8] and 55
successfully implemented in other disease models, enabling the ad hoc development of 56
diverse phenotyping concepts [9 - 12]. "

Author Response

We thank the reviewer for the comment that have helped us to improve the paper. All changes have been indicated by red color within the text. Below you will find a point-by-point response to your comments.

Comments 1. The article is well-written and interesting, giving statistical validation to clinical impressions of first-line healthcare workers in pandemia.

I will just suggest to turn this statement impersonal and focus more on explanation of the methodology. "We were the first to phenotype comorbidity in obstructive sleep apnea (OSA) using a 52 combination of preclustering principal component analysis to identify latent structures 53 within our datasets, and subsequently map them using the Two-Step Cluster algorithm 54 [3]. The methodology we proposed has been adopted by other research groups [4 - 8] and 55 successfully implemented in other disease models, enabling the ad hoc development of 56 diverse phenotyping concepts [9 - 12]. "

Response: Thank you for your comment and the appraisal of our work. Following the reviewer’s suggestion, we have amended the statement and expanded on the contributions of pattern recognition and this particular technique in clinical research:

This methodology we proposed has been adopted by other research groups [4 - 8] in obstructive sleep apnea, revealing that i.e. severity based definitions could ignore non-linear relationships between disease characteristics and severity indices [3], as well as omit specific relationships (i.e. latent structures) between clinical manifestations and laboratory findings [4-6]. Other implementations in sleep disordered breathing have shown that aside from classical definitions, i.e. based on an index of morbidity, different combinations of symptoms or laboratory findings could be used as phenotyping variables [7,8]. Aside from sleep disordered breathing, this proposed methodology [3] and successfully has been used successfully in other disease models. implemented in other disease models, enabling the ad hoc development of diverse phenotyping concepts in i.e. idiopathic Parkinson’s disease, obesity, venous thromboembolism [9 -– 12]. Furthermore, data from multicenter studies have also been shown pro-duce biologically relevant phenotypes via our proposed methodology [13]. These studies have shown that a data-driven, pattern recognition approach is both flexible, robust and allows the discovery of phenotypes that are independent of clinical preconceptions, often subjects to salience bias.

Round 2

Reviewer 1 Report

Dear Authors,
In light of the straightforward responses to my concerns, I suggest acceptance of the manuscript
Best regards